# Estimation of Carbon Stocks of Birch Forests on Abandoned Arable Lands in the Cis-Ural Using Unmanned Aerial Vehicle-Mounted LiDAR Camera

Nikolay Fedorov [1,2,*], Ilnur Bikbaev [1,2], Pavel Shirokikh [1,2,*], Svetlana Zhigunova [1,2], Ilshat Tuktamyshev [1,2], Oksana Mikhaylenko [2], Vasiliy Martynenko [1,2], Aleksey Kulagin [1,2], Raphak Giniyatullin [1,2], Ruslan Urazgildin [1,2], Mikhail Komissarov [1,2] and Larisa Belan [2]

1　Ufa Institute of Biology, Ufa Federal Research Centre of the Russian Academy of Sciences, 450054 Ufa, Russia; ilnur.bikbaev.90@mail.ru (I.B.); zigusvet@yandex.ru (S.Z.); ishatik@yandex.ru (I.T.); vb-mart@mail.ru (V.M.); coolagin@list.ru (A.K.); grafak2012@yandex.ru (R.G.); urv@anrb.ru (R.U.); mkomissarov@list.ru (M.K.)

2　Laboratory of Climate Change Monitoring and Carbon Ecosystems Balance, Ufa State Petroleum Technological University, 450064 Ufa, Russia; trioksan@mail.ru (O.M.); belan77767@mail.ru (L.B.)

*　Correspondence: fedorov@anrb.ru (N.F.); shirpa@mail.ru (P.S.); Tel.: +7-8(937)-353-38-96 (N.F.); +7-8(927)-334-03-60 (P.S.)

**Abstract:** Currently, studies investigating the carbon balance in forest ecosystems are particularly relevant due to the global increase in $CO_2$ content in the atmosphere. Due to natural reforestation over the past 25–30 years, birch (*Betula pendula* Roth.) forests were extensively grown and established on abandoned agricultural lands in Bashkir Cis-Ural (Republic of Bashkortostan, Russia). The significant positive aspect of reforestation on fallow lands is the carbon sequestration that takes place in the tree phytomass, especially at the growth stage of stand formation. The aim of this article is to test the approach of using a UAV-mounted LiDAR camera to estimate the phytomass and carbon stocks in different-aged birch forests growing on abandoned arable lands in Bashkir Cis-Ural. The methodology was developed using 28 sample plots, where the LiDAR survey was performed using a DJI Matrice 300 RTK UAV. Simultaneously, the stand characteristics and phytomass of stem wood were also estimated, using traditional methods in the field of forest science. The regression equations of phytomass dependence on stand characteristics at different stages of reforestation were constructed using data obtained from LiDAR imagery. It was shown that the above-ground tree biomass could be precisely estimated using the index obtained by multiplying the number of trees and their average height. A comparison of the data obtained using traditional and LiDAR survey methods found that the accuracy of the latter increased in conjunction with stand density. The accuracy of estimation ranged from 0.2 to 6.8% in birch forests aged 20 years and over. To calculate carbon stocks of the above-ground tree stands, the use of regional conversion coefficients is suggested, which could also be applied for the estimation of carbon content in trunk wood and leaves. An equation for the calculation of above-ground biomass carbon stocks of birch forests on abandoned arable lands is proposed.

**Keywords:** 3D laser scanning; above-ground biomass; *Betula pendula*; carbon sequestration; regression models





## 1. Introduction

In the second half of the 20th century, abandoned agricultural lands increased significantly in North America, the former Soviet Union and South Asia, and subsequently Europe, South America and China, due to the reduction in agriculture intensity [1–8]. An analysis of the History Database of Global Environment 3.0 (HYDE) database of Campbell, Lobell and Field [9] showed that 269 million ha of croplands and 479 million ha of pastures were abandoned globally over the last three centuries. However, after accounting for forest

regrowth and urbanization, the total area of abandoned agricultural lands ranged from 385 to 472 million ha [10]. Many reasons led to the abandonment of croplands, such as the dismantling of the state administration system, the introduction of free market principles, the end of state regulation and support, and land reforms [11].

Russia ranks first in the world in terms of lands excluded from agricultural use, which, according to various estimates, ranges from 76 to 97.2 million ha [12,13]. In the last 20–30 years, many areas of abandoned cropland have been actively reforested. In most cases, abandoned croplands have become overgrown with fast-growing tree species with high seed production. The species diversity of trees on abandoned arable lands depends mainly on the availability of seed sources, the species composition of the adjacent forest stands, soil fertility, and the utilization regime before and after cessation of plowing [14,15].

Among the regions of Russia, the Republic of Bashkortostan has the highest percentage of unused arable agricultural lands overgrown with forest vegetation, the area of which is more than 4 million ha [16]. In different districts of Bashkortostan, overgrowth occurs in terms of different tree species, including the Scots pine (*Pinus sylvestris* L.), silver birch (*Betula pendula* Roth.), and European aspen (*Populus tremula* L.), and less frequently, the European oak (*Quercus robur* L.) and Norway maple (*Acer platanoides* L.). In some cases, an overgrowth of Box Elder (*Acer negundo* L.) is observed [17]. However, the silver birch is one of the main forest-forming species on fallow lands and rapidly occupies treeless areas after the abandonment of agricultural activities, due to its strong juvenile growth, abundant seed production, and large soil seed bank [18,19]. Over the last three decades, forest plantations dominated by this species have formed on large areas of abandoned arable lands in Cis-Ural. In the region, the average carbon stocks in 25–30-year-old birch tree stands on abandoned agricultural land is 74.9 t/ha [20], which is close to that in Scandinavia for a 32-year-old birch tree stand (87.7 t/ha) [21]. Silver birch also demonstrates a wide natural distribution on the European part of the continent [22], and abandoned arable lands overgrown with this species can also be identified in Poland [23], Estonia [24–26], Sweden [27], and other European countries.

The positive aspect of reforestation on fallow lands is carbon sequestration that 76 takes place in tree phytomass, especially at the stages of stand formation [28–30]. Increased plant biomass after the cessation of plowing contributes to increased organic matter content by integrating plant litter and roots, thereby increasing soil organic carbon content [31–34]. Thus, abandoned agricultural lands are large carbon stores and contribute to the reduction in greenhouse gas emissions; thus, playing an important role in the global climate change processes [35–37].

The increasing succession of silver birch on abandoned farmlands and the emerging need to manage these areas has necessitated research into the structure of birch forests [38]. Mapping of the above-ground woody biomass (AGB) on abandoned agricultural land is required in terms of the relevant stakeholders to monitor the spatial dynamics of farmland afforestation, assess carbon sequestration, and establish appropriate natural resource management [39,40].

Trunk diameter at breast height, individual tree height, and crown base height are typically measured for stand inventories and forest management. Although traditional field measurements are still widely practiced, they have some disadvantages because they are time-consuming and limited in terms of spatial scale. Since the early 2000s, LiDAR has been used worldwide as an alternative to traditional forest inventory methods of observations. Unmanned aerial vehicle (UAV) surveys provide objective and maximally accurate data concerning forest stands, including tree heights, crown diameters, and volume, and the number of trees per unit area [23,41–49]. Tree height estimation using a laser approximated more closely to the real height of felled trees than traditional field measurements [50]. Since the literature indicates a reasonably high efficiency in using LiDAR imagery for operational applications in simple forest structures (e.g., single-tier stands) [51,52], we hypothesized that this approach would be applicable to the study of secondary reforestation successions on abandoned farmlands.



This research aimed to test a method using a UAV-mounted LiDAR camera to estimate the phytomass and carbon stocks of birch forests of different ages growing on abandoned arable lands in Bashkir Cis-Ural.

## 2. Materials and Methods

The research was conducted in Cis-Ural on the "Mishkinsky carbon polygon" (Mishkinsky district, Republic of Bashkortostan, Russia). The climate in the study area is temperate, continental, and relatively humid (Dfb according to Köppen–Geiger climate classification). The annual average air temperature is +3.8 °C and precipitation is 589 mm. The soil cover consists mostly of grey forest soil and less frequently of dark grey forest soils (to determine soil type and its morphological properties, the soil profiles were excavated on each sample plot).

The method for estimating phytomass and carbon stocks of forest vegetation was conducted on 28 sample plots (sized 30 × 30 m) located on abandoned arable land overgrown with silver birch (Figure 1).

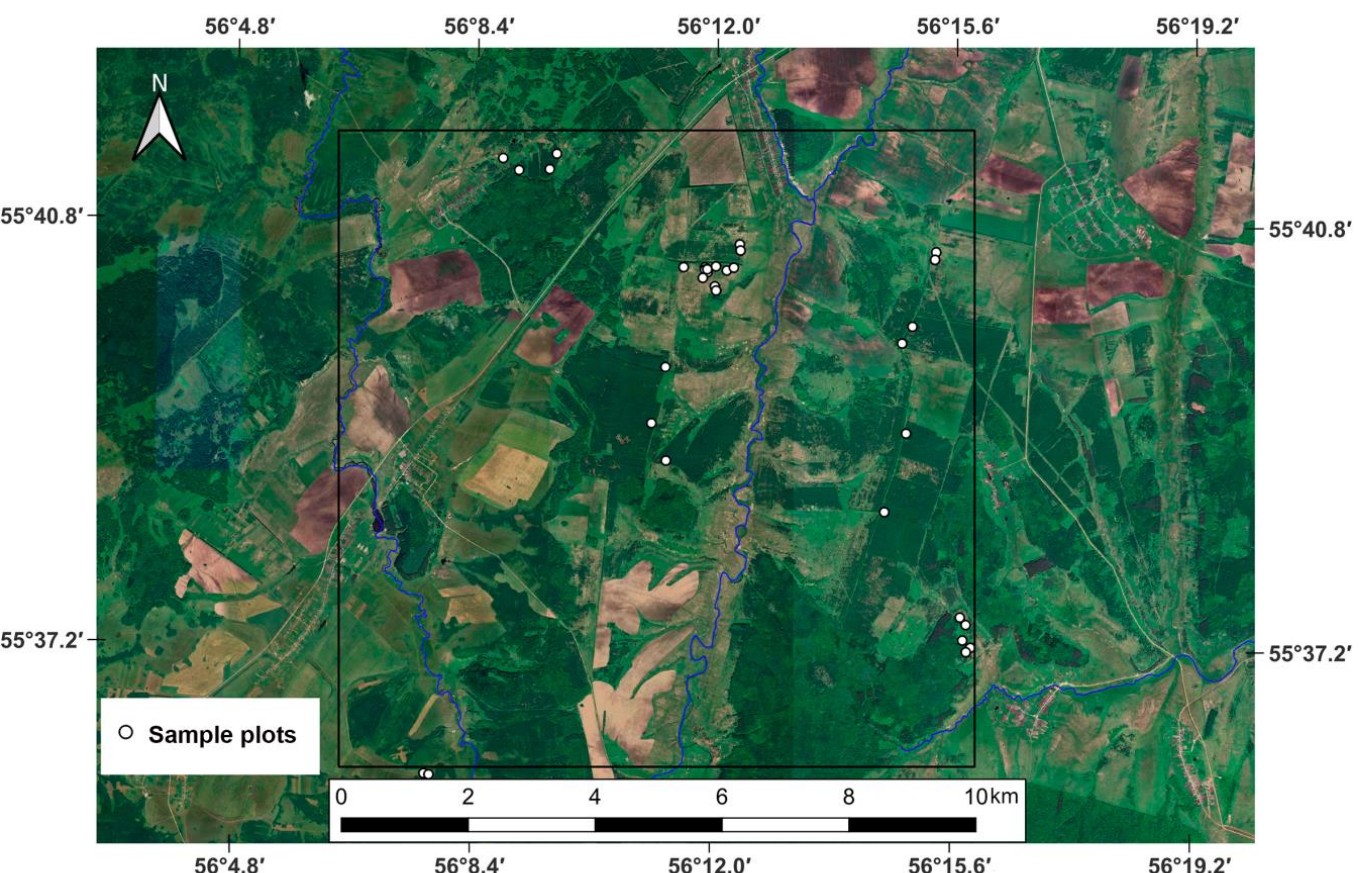

**Figure 1.** Location of sample plots in the Mishkinsky carbon polygon.

The sample plots for carbon stock calculations were selected and identified using traditional field reconnaissance and LiDAR surveys. The sample plots were distinguished according to the stages of reforestation. Five main stages of natural reforestation were identified, within which two variants with different stand and crown density (variant 1 with relatively low coverage, and variant 2 with higher coverage) were distinguished (Table 1). At stage I, only variant 1 with two sample plots was found, but there were no birch forests belonging to stage V, which has a sparse tree layer.

The above-ground biomass and the productivity of birch stands were estimated at the end of summer (August) when the biomass formation processes were completed. The model tree method was used to estimate the timber stock. In order to obtain the results

of the phytomass reserves in the trees stands, we were guided by the generally accepted methods of sampling plots and taking an inventory of tree stands. The biomass of living trees, dead, and fallen wood was estimated using the data of the number of trees, their diameter at breast height (DBH, i.e., 1.3 m height above the ground), and information about the selected model trees within each sample plot [53,54]. Samples of biomass were crushed with cutting mills of the VLM series (Vilitek LLC, Moscow, Russia) to a particle size of less than 0.5 mm. The carbon content in the samples was determined using a CHNS EA-3100 elemental analyzer (Eurovector, Pavia, Italy). Conversion coefficients were calculated from the carbon content of the samples to estimate carbon stocks in stem wood and leaves [20].

**Table 1.** The characteristics of *Betula pendula* tree stands at different stages of reforestation in the study area.

| Stage of Reforestation | Trees Height, m | Age of Trees, Years | Diameter of Trunks, cm | Variant 1 (Projective Coverage, %) | Number of Plots Variant 1 | Variant 2 (Projective Coverage, %) | Number of Plots Variant 2 |
|---|---|---|---|---|---|---|---|
| I | 0.5–1.5 | 3–8 | – | 1–5 | 2 | 7–10 | – |
| II | 2–3 | 9–14 | 1–3 | 10–20 | 1 | 30–50 | 5 |
| III | 5–8 | 15–20 | 6–8 | 30–50 | 3 | 60–80 | 1 |
| IV | 9–14 | 20–25 | 10–14 | 50–60 | 4 | 75–90 | 3 |
| V | 15–18 | 25–30 | 16–20 | 50–60 | – | 75–90 | 6 |

Laser scanning of tree stands was conducted using a DJI Matrice 300 RTK UAV equipped with a Zenmuse L1 LiDAR camera (SZ DJI Technology Co., Shenzhen, China) from a 100 m altitude. The Zenmuse Lidar L1 combines a Livox Lidar module, a high-precision IMU and a 1-inch CMOS camera on a 3-axis stabilized system (covering up to 2 km$^2$ in a single span, with a vertical accuracy of 5 cm, a horizontal accuracy of 10 cm, a point rate of 240,000 pts/s, and a detection range of 450 m). In the preparatory phase, a flight mission was planned—the survey area was set in KML format, which was loaded into the DJI Pilot 2 UAV's control panel software (v.2). After the flight plan was uploaded, the drone performed image acquisition in automatic mode. The images were saved in JPG format and, in parallel, the georeferenced coordinates of the images were recorded using the D-RTK2 mobile station. The resulting images were imported into the DJI Terra program and stitched together automatically. The 3D point cloud in the LAS format with a density ranging from 300 to 950 pcs/m$^2$ was formed. The post-processing of the stitched images was performed in the program LiDAR 360 Version 4.5 (Green Valley International, Berkeley, CA, USA) [55,56]. In the first stage, each UAV flyover section was cleared of points located outside the main scanning area ("outliers"). Next, the set of points was classified and divided into two types: points of the Earth's surface and other points above them. A digital elevation model (DEM) was built using the points of the first type. The remaining points were used to create a digital terrain model (DTM), which contained spatial information about the surface position of all objects in the areas above the Earth's surface. By excluding DEM data from the DEM, a digital forest canopy model (DFM) was created, which is an image of the crowns of trees. The pixel size of the resulting images was 4.0 cm. The number and height of trees, their geometric coordinates on the image, crown diameters, their areas and volumes were calculated in the ASL Forest module of the LiDAR 360 program, which was required to obtain digital forest canopy models that represent the image of tree crowns (Figure 2).

Regression equations of the dependence of phytomass and carbon stocks on stand characteristics obtained by the LiDAR survey were constructed. Regression analysis was carried out in "Statgraphics centurion XV". The "Comparison of Alternative Models" algorithm was used to select the optimal regression models [57]. The coefficients of the correlation (R) and determination (R$^2$) and the estimation standard error (ESE) were used as model quality criteria. The R$^2$ coefficient reflects the proportion of the variance of the dependent variable explained by the model under consideration. The ESE is a measure of the mean error variance, the difference between the indicator values predicted by the

regression model and the indicator values in the sample. In other words, the standard error of regression is the average distance by which the observed values deviate from the regression line.

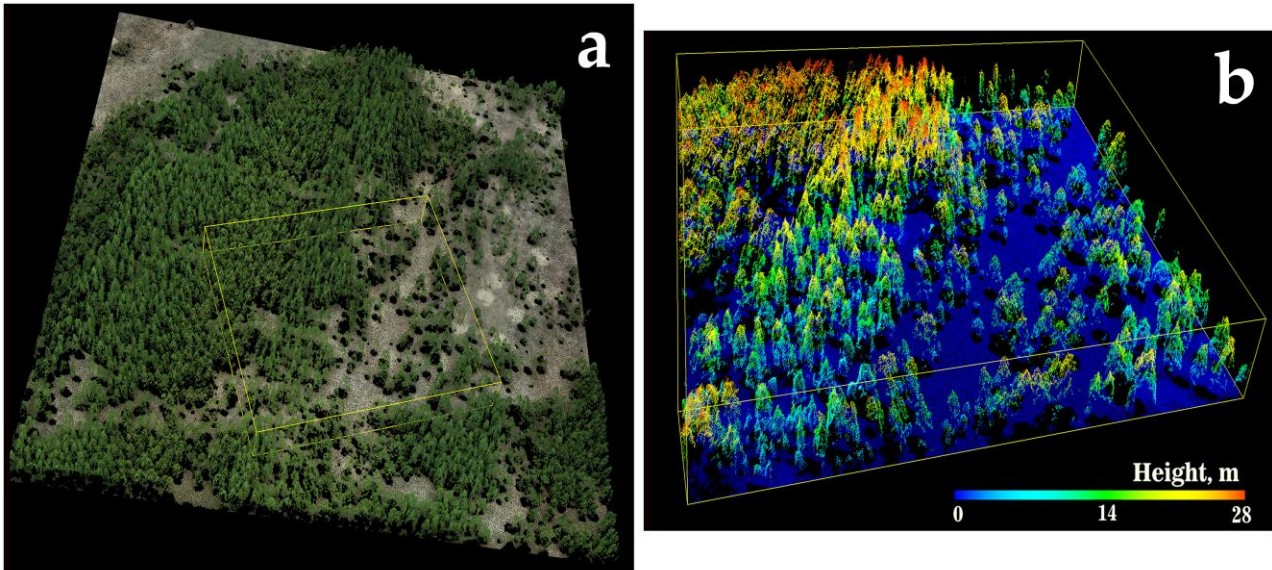

**Figure 2.** View on the sample plot (No 1190, Stage V, Variant 2) during LiDAR survey (**a**) and 3D model of the stand crowns (**b**).

## 3. Results

Table 2 shows the results of the regression models of the phytomass of stem wood with branches and the phytomass of leaves, dependent on the stand characteristics calculated from the results of the LiDAR survey. All models are described using nonlinear equations.

The values of R between the phytomass of stem wood with branches and stand characteristics calculated from the LiDAR survey (for almost all of the cases) are greater than 0.90 for all cases, and $R^2$ is greater than 88%.

**Table 2.** Regression models for calculating phytomass of stem wood with branches and phytomass of leaves based on stand characteristics calculated from LiDAR survey results.

| Stand Characteristics | Regression Model Equation | R | $R^2$ | ESE |
|---|---|---|---|---|
| **Phytomass of stem wood with branches** | | | | |
| Multiplying the sum of tree crowns diameters by their average height | Square root-Y model: $Y = (6.25464 + 0.0320395 \times X)^2$ | 0.98 | 95.5 | 9.8 |
| Multiplying the number of trees by their average height | Square root-Y model: $Y = (3.33595 + 0.1372 \times X)^2$ | 0.97 | 94.4 | 10.9 |
| Multiplying the sum of tree crown area by their average height | Square root-Y model: $Y = (7.46822 + 0.00848049 \times X)^2$ | 0.97 | 93.5 | 11.7 |
| Trees average height | Square root-Y squared-X model: $Y = (9.49088 + 0.430736 \times X^2)^2$ | 0.96 | 92.1 | 12.9 |
| Sum of trees crown diameters | Logarithmic-Y square root-X model: $Y = \exp(0.875185 + 0.576223 \times \sqrt{X})$ | 0.94 | 89.2 | 1.2 |
| Multiplying the sum of tree crown volumes by their average height | Double square root model: $Y = (3.98812 + 0.486332 \times \sqrt{X})^2$ | 0.94 | 88.8 | 15.4 |
| Sum of tree crown areas | Logarithmic-Y square root-X model: $Y = \exp(1.64388 + 0.270862 \times \sqrt{X})$ | 0.92 | 84.0 | 1.5 |
| Sum of tree crown volumes | Logarithmic-Y square root-X model: $Y = \exp(2.16932 + 0.138317 \times \sqrt{X})$ | 0.88 | 78.0 | 1.7 |

**Table 2.** *Cont.*

| Stand Characteristics | Regression Model Equation | R | R$^2$ | ESE |
|---|---|---|---|---|
| **Phytomass of leaves** | | | | |
| Multiplying the sum of tree crown diameters by their average height | Double square root model:<br>Y = (1.98201 + 0.3495 × √X)$^2$ | 0.92 | 85.4 | 3.4 |
| Multiplying the number of trees by their average height | Square root-Y model:<br>Y = (3.38375 + 0.0254409 × X)$^2$ | 0.92 | 84.4 | 3.5 |
| Sum of tree crown diameters | Logarithmic-Y square root-X model:<br>Y = exp(0.656103 + 0.377484 × √X) | 0.91 | 83.6 | 1.0 |
| Multiplying the sum of tree crown areas by their average height | Double square root model:<br>Y = (2.79809 + 0.172115 × √X)$^2$ | 0.91 | 82.3 | 3.8 |
| Tree average height | Square root-Y model:<br>Y = (1.32037 + 1.37447 × X)$^2$ | 0.90 | 81.0 | 3.9 |
| Sum of tree crown areas | Double square root model:<br>Y = (1.70192 + 0.641025 × √X)$^2$ | 0.88 | 78.2 | 4.2 |
| Multiplying the sum of tree crown volumes by their average height | Double square root model:<br>Y = (3.70018 + 0.0880252 × √X)$^2$ | 0.87 | 76.0 | 4.4 |
| Sum of tree crown volumes | Double square root model:<br>Y = (2.91063 + 0.328646 × √X)$^2$ | 0.86 | 73.2 | 4.7 |

Note: R—correlation coefficient; R$^2$—coefficient of determination; ESE—estimation standard error.

Figure 3 shows the above-described regression equations used to calculate the phytomass of stem wood with branches from the stand characteristics obtained from the LiDAR survey. In the graphs, the green lines indicate confidence intervals for the mean response of the variable, which reflect how accurately the position of the line was estimated with the existing data sample. The gray lines on the graphs indicate the prediction limits for new observations, allowing us to describe how accurately we can predict where the new observations will lie, which will vary around the true line with a standard deviation. Table 2 and Figure 3 show that regression models built via the multiplication of the sum of tree crown diameters by their average height, as well as by the multiplication of the number of trees by their average height are optimal for calculating the phytomass of stem wood with branches.

Separate regression models were utilized to calculate the phytomass of leaves. Table 2 and Figure 4 show that all models are described using nonlinear equations as well as stem wood models. The values of R between the phytomass of the leaves and stand characteristics calculated via the LiDAR survey are (for almost all of the cases) greater than 0.86, and R$^2$ is greater than 73%. Thus, the calculated regression equations used to estimate the phytomass of leaves from stand characteristics obtained using LiDAR imagery are quite close in accuracy to the estimates of stem wood with branches. The phytomass of leaves, as in the case with the the phytomass of stem wood with branches, is most accurately estimated via the multiplication of the sum of tree crown diameters by their average height and via the multiplication of the number of trees by their average height (Table 2).

Figure 4 shows that the phytomass of leaves has more variability than the phytomass of stem wood with branches due to changes in the crown volume at the same tree height; this change is caused by stand projective cover. However, since most carbon storage occurs in the stem wood with branches, this variability does not play a major role in calculating the total above-ground carbon stocks in the stand.

The previously calculated conversion coefficients were used to estimate the carbon stocks in the above-ground phytomass of the tree stands [58]. The carbon content in the phytomass of birch stem wood with branches within the studied polygon is fairly constant regardless of tree age and averages at 48.5%. The variability in this indicator at different sample plots did not exceed 3% [59]. Therefore, in order to obtain the carbon stock in stem wood with branches, it is sufficient to multiply the phytomass value obtained using one of the regression equations by 0.485. Similarly, the mean carbon content of the leaves is 48.8%,

with a maximum variation from 46.3 to 51.6%. Therefore, the carbon stocks in leaves were calculated by multiplying the phytomass by 0.488.

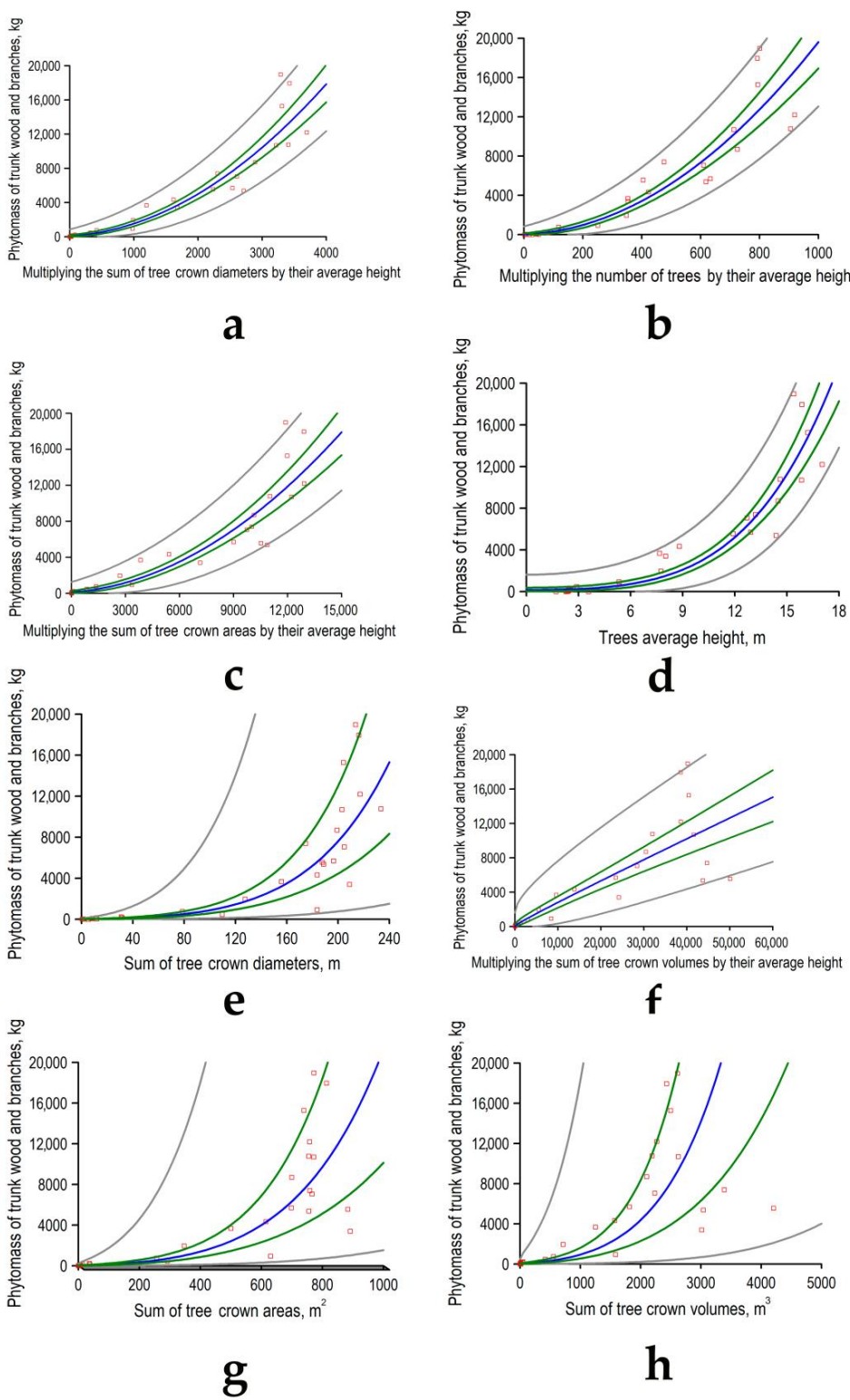

**Figure 3.** Dependence of phytomass of stem wood with branches on the index obtained from the LiDAR data (**a**) multiplying the sum of tree crown diameters by their average height; (**b**) multiplying the number of trees by their average height; (**c**) multiplying the sum of trees crown areas by their average height; (**d**) tree average height; (**e**) sum of trees crown diameters; (**f**) multiplying the sum of tree crown volumes by their average height; (**g**) sum of tree crown area; (**h**) sum of tree crown volumes.

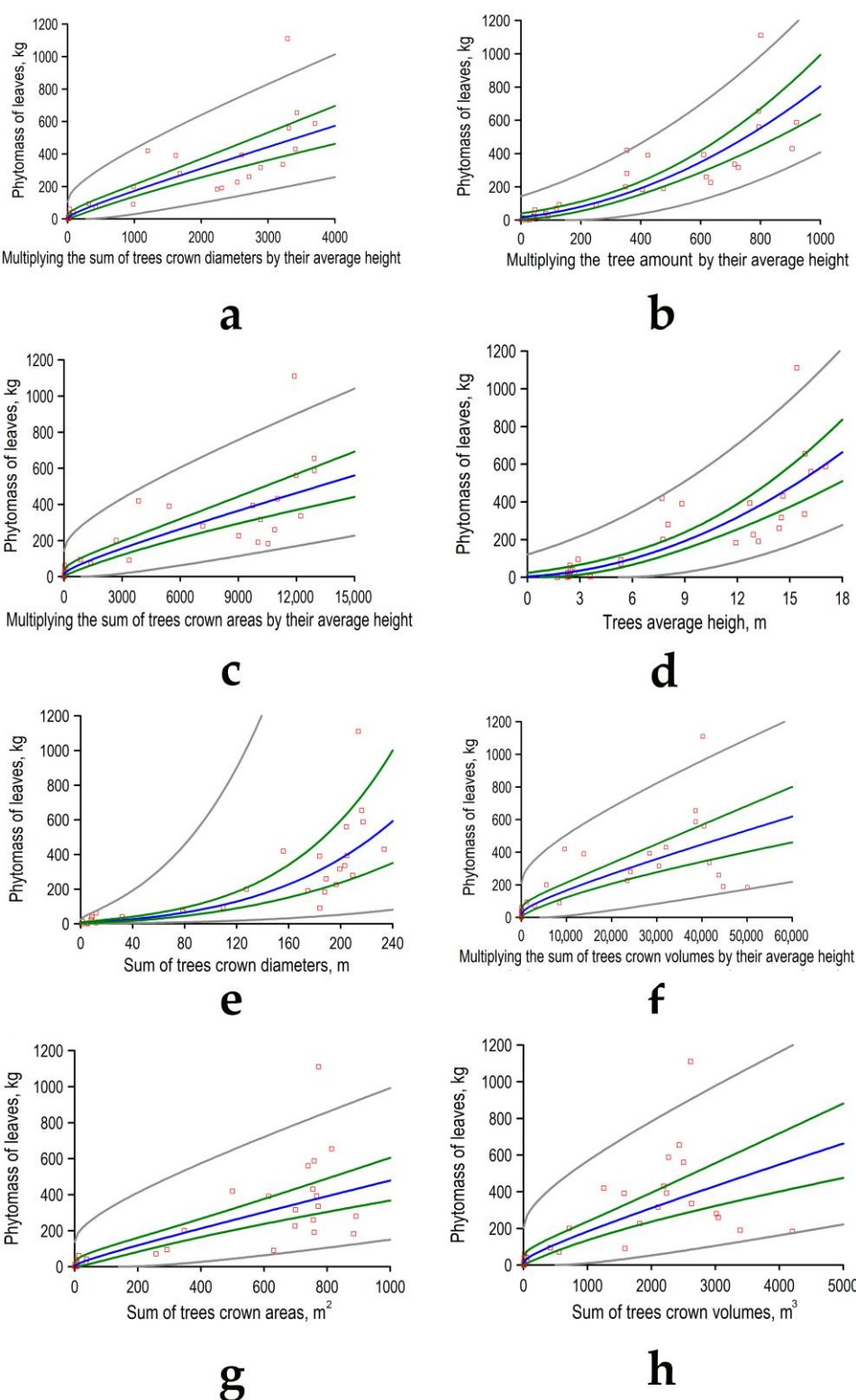

**Figure 4.** Dependence of phytomass of leaves on the index obtained from the LiDAR data (**a**) multiplying the sum of trees crown diameters by their average height; (**b**) multiplying the number of trees by their average height; (**c**) multiplying the sum of trees crown areas by their average height; (**d**) tree average height; (**e**) sum of tree crown diameters; (**f**) multiplying the sum of trees crown volumes by their average height; (**g**) sum of trees crown areas; (**h**) sum of tree crown volumes.

Furthermore, the difference between the values of carbon stocks in the above-ground part of the stand (including stem wood with branches and leaves) calculated from the LiDAR survey data and the values obtained when using the traditional method using a model tree was assessed (Table 3). The phytomass of the leaves and phytomass of stem wood with branches were calculated in each case using the same parameters (e.g., the multiplication of the number of trees by their average height, average height, etc.).

**Table 3.** Carbon stocks in the above-ground part of the forest stand on abandoned arable lands in the Cis-Ural area overgrown with silver Birch calculated using the traditional method and using different indicators obtained from the LiDAR data.

| Stage (S) and Variant (V) of Overgrowth | | | | | | | |
|---|---|---|---|---|---|---|---|
| S1V1 | S2V1 | S2V2 | S3V1 | S3V2 | S4V1 | S4V2 | S5V2 |
| **Carbon stocks in above-ground biomass of tree layer based on traditional field measurement, kg/ha \*** | | | | | | | |
| 24.7 ±15.2 | 299.2 ±136.5 | 1410.5 ±379.4 | 10,579.4 ±4581.9 | 19,431.4 | 26,726.6 ±5204.4 | 33,781.2 ±2291.9 | 77,554.0 ±8589.4 |
| **Carbon stocks in above-ground biomass of tree layer based on LiDAR data, kg/ha** | | | | | | | |
| **By multiplying the sum of trees crown diameters and their average height** | | | | | | | |
| 278.1 ±27.1 (1023.5) | 302.7 ±18.5 (1.2) | 711.3 ±243.0 (−49.6) | 7519.6 ±2187.0 (−28.9) | 26,308.7 (35.4) | 28,233.4 ±7708.9 (5.6) | 40,836.7 ±2720.3 (20.9) | 71,634.1 ±3841.3 (−7.6) |
| **By multiplying the number of trees and their average height** | | | | | | | |
| 196.5 ±45.4 (694.0) | 447.3 ±98.3 (49.5) | 1298.8 ±327.4 (−7.9) | 8564.0 ±3066.8 (−19.0) | 15270.5 (−21.4) | 26,680.0 ±5377.8 (−0.2) | 36,091.6 ±6731.6 (6.8) | 76,492.4 ±5032.5 (−1.4) |
| **By multiplying the sum of tree crown areas and their average height** | | | | | | | |
| 369.6 ±17.2 (1393.6) | 366.5 ±5.4 (22.5) | 641.8 ±189.2 (−54.5) | 6619.4 ±1762.6 (−37.4) | 26,631.0 (37.1) | 31,554.6 ±10467.7 (18.1) | 46,109.7 ±3009.1 (36.5) | 65,251.1 ±3981.3 (−15.9) |
| **Based on average tree height** | | | | | | | |
| 1059.6 ±258.8 (4181.4) | 880.0 ±9.7 (194.1) | 951.6 ±34.5 (−32.5) | 4403.7 ±1182.2 (−58.4) | 26,308.7 (35.4) | 28,412.1 ±9950.9 (6.3) | 33,896.3 ±2241.2 (0.3) | 74,375.6 ±6267.3 (−4.1) |
| **Based on the sum of tree crown diameters** | | | | | | | |
| 52.8 ±14.8 (113.2) | 85.5 ±17.0 (−71.4) | 1396.2 ±1016.0 (−1.0) | 18,130.1 ±7350.6 (71.4) | 26,308.7 (35.4) | 27,004.2 ±5356.4 (1.0) | 43,901.0 ±3578.2 (30.0) | 62,994.7 ±5616.1 (−18.8) |
| **By multiplying the sum of tree crown volumes and their average height** | | | | | | | |
| 285.1 ±78.1 (1052.2) | 244.7 ±18.1 (−18.2) | 868.4 ±389.6 (−38.4) | 11,368.6 ±2471.3 (7.5) | 26,308.7 (35.4) | 39,368.9 ±12,250.0 (47.3) | 49,310.2 ±9213.0 (46.0) | 53,160.7 ±2234.8 (−31.5) |
| **Based on the sum of tree crown areas** | | | | | | | |
| 87.4 ±26.4 (253.3) | 94.5 ±11.3 (−68.4) | 922.1 ±627.4 (−34.6) | 14,299.1 ±5619.1 (35.2) | 26,308.7 (35.4) | 32,552.9 ±9417.0 (21.8) | 59,917.2 ±12695.8 (77.4) | 50,500.3 ±3386.7 (−34.9) |
| **Based on the sum of tree crown volumes** | | | | | | | |
| 136.0 ±24.8 (449.4) | 129.8 ±7.6 (−56.6) | 423.3 ±195.6 (−70.0) | 7423.4 ±2628.4 (−29.8) | 26,308.7 (35.4) | 66,205.1 ±30,663.3 (147.7) | 142,022.3 ±94,361.9 (320.4) | 41,472.7 ±4110.0 (−46.5) |

Note: the difference between the measurement results is provided in brackets, %. \* More details in the publication [58].

Table 3 shows that as stand projective cover increases, the accuracy of LiDAR estimates increases. The best model for calculating the above-ground biomass of birch forests is the model that multiplication the number of trees by their average height:

$$AGB = (3.33595 + 0.1372 \times X)^2 + (3.38375 + 0.0254409 \times X)^2 \qquad (1)$$

where AGB represents the above-ground biomass of a birch stand, and X—multiplication of the number of trees by their average height.

Using conversion coefficients, carbon stocks in the above-ground part of birch forests can be calculated according to the formula:

$$CS = (3.33595 + 0.1372 \times X)^2 \times 0.485 + (3.38375 + 0.0254409 \times X)^2 \times 0.488 \qquad (2)$$

where CS represents the carbon stock in the birch stand, and X—multiplication of the number of trees by their average height.

The accuracy of carbon stock estimates using this formula in birch forests that are of 20 years of age and older ranges from 0.2 to 6.8%, and in birch forests of 9–20 years old with projective cover greater than 30%, it ranges from 7.9 to 21.4% (Table 3). At stage I of overgrowth, there are very strong discrepancies between the results of calculations using the traditional method and LiDAR data. However, the values at this stage are on average only 24.7 kg of carbon per ha, and unlike the other stages, the main carbon storage occurs not in the tree layer but in the herb layer.

## 4. Discussion

Our results coincide with the literature data in providing a sufficiently accurate estimation of the phytomass of stands of trees with high projective cover and their carbon stocks using LiDAR imagery [56]. The relationship between stand characteristics obtained using LiDAR and biomass calculated via traditional field measurements are, in all cases, described by using nonlinear equations, which is also consistent with the literature data [58]. Comparing the results of carbon stock estimation in the above-ground part of the stand using the traditional methods and LiDAR survey data, we found that the best model for calculating the phytomass of stem wood with branches, as well as the phytomass of leaves, is the model of multiplication of the number of trees by their average height. The tree height included in the calculation formula is a key parameter whose relationship to above-ground biomass has also been reported in other studies [59]. In our investigation, the average height of trees is important due to its correlation with the stages of reforestation, differing according to the age and the above-ground biomass of trees. However, in stands with low projective cover, height is less important because at the same height, depending on stand density, crown size can vary greatly. Therefore, at stage III of overgrowth (tree heights of 5–8 m and a crown densities of 40–65%), multiplication of the number of trees by the average stand height and multiplication of the sum of diameters by the average stand height provide more accurate estimates of biomass and carbon stocks than average stand height on its own. The results explained more than 94% of the variance, which is very good and consistent with the accuracy of the model using the LiDAR Biomass Index (LBI) calculated from the estimated crown volume of individual trees [60]. The choice of this indicator is also justified, as among the five main forest stand parameters (number of trees, tree average height, sum of tree diameters, area, and volumes of tree crown), only the number of trees and their average height are independent. The sum of crown diameters and the sum of crown areas depend on the average height of the trees, which increases with age, and with the number of trees. The sum of tree crown volumes also depends on the average height of the trees because it is related to the age of the stand. Therefore, the three parameters (multiplication of the sum of tree crown diameters by their average height, multiplication of the sum of tree crown areas by their average height, and multiplication of the sum of tree crown volumes by their average height) may provide an inaccurate estimate because the formula includes dependent variables. Thus, the multiplication of the number of trees by

their average height is the most appropriate. The proposed method for calculating carbon stocks in the above-ground part of the stand on abandoned agricultural lands using the multiplication of the number of trees by the average height provides sufficiently accurate estimates for birch forests that are more than 9 years old with projective cover of more than 30%.

As noted above, as the density of forest stand increases, the accuracy of the LiDAR survey estimates also increases. The greater discrepancy between the results of carbon stock calculation using traditional methods and LiDAR methods in sparse stands is due to the fact that in stages II-III of reforestation, trees of the same age may have strong differences in the size and shape of their crowns. The influence of stand density on the living crown ratio in silver birch stands has also been revealed in other studies [61]. This coincides with the findings in the literature showing that the relative deviations of UAV estimates increased in those stands consisting of isolated groups of trees, indicating the potential limitations of the approach and the need for its further development [2,62].

The strongest discrepancies are at stage I of reforestation. This is explained by the fact that the birch stand of this stage is represented by 3–5-year-old trees with heights ranging from 0.5 to 1.5 m, which grow mosaically and may be inadequately distinguished from shrubs and large herbaceous plants on LiDAR images. This may be the reason for the significant overestimation of results. At this stage, the main carbon storage is not in the woody understory but in the herbaceous layer. Thus, the plant communities of stage I and the sparse variant 1 of stage II of reforestation should not be classified as forests. At the initial stage of reforestation succession, the main carbon storage occurs in the herbaceous layer. Probably, in this case, it will be more effective to use regression equations based on vegetation indices calculated from high-resolution multispectral images from UAVs or space satellites [63].

In addition, the differences between biomass estimated by using traditional measurements and using LiDAR data may be explained by the fact that the centers of the sample plots were marked using a GPS navigator with an accuracy of 3 m. The UAV imagery may have shifted the boundaries of the sample plots, which increased the influence of the mosaic distribution of birch trees in the early stages of reforestation. However, birch forests younger than 15 years old occupy insignificant areas in the Cis-Ural; therefore, the method can be used to estimate carbon stocks in birch forests in the region.

In the Cis-Ural, abandoned arable lands are overgrown not only with silver birch but also with Scots pine, which dominates in the close-quartered presence of its stands. The growth rate of pine is much slower than that of birch; so in the early stages of reforestation, pine forests with sparse stands are more common than in the early stages of birch forests. In this regard, the method of calculating the biomass of individual trees 341 using the LiDAR biomass index (LBI) can be used for the above-ground biomass of sparse pine forest stands, which takes into account the crown area of model trees at different heights [60]. In forests with high projective cover, the use of the latter method is technically difficult. It is worth noting that during LiDAR imagery analyses, the ravines, temporary watercourses, and places of sediment accumulation were easily detected; thus, LiDAR survey data could also be potentially used to identify soil erosion processes, and this requires further research in the study region. In other natural and climatic conditions, this method has already shown its effectiveness [64,65].

Given the high prevalence of abandoned agricultural lands overgrown with birch trees, the LiDAR remote sensing approach has great potential in the determination of carbon sequestration and greenhouse gas balance in the ecosystems [66]. The investigations and mapping changes in live above-ground biomasses in space and time using LiDAR can provide critical insights into the drivers of carbon flux and ecological change during forest community succession [67–69] and will enable predictions of future soil–reforestation interactions under climalic and land use transformations [70].

## 5. Conclusions

Since the estimation of carbon stocks in birch forests older than 9 years (stages III–V of reforestation) is quite accurate, our proposed regression equations can be used to analyze the dynamics of biomass and carbon stocks in stands across large areas, both in the Cis-Ural and in other regions. The conversion coefficients for converting biomass to carbon stocks in other regions need to be verified and, if necessary, refined. Birch succession on abandoned agricultural land necessitates the management of these areas, including the selection of their future use. In the cases of low productivity or low density in terms of the trees, the overgrown lands can be used as hayfields and pastures or even as arable land (after clearing). Highly productive birch forests require silvicultural measures, such as thinning at high stand densities. Such communities can be used as timber sources and as carbon farms established for carbon sequestration. In the latter case, additional activities will be required to increase the rate of carbon sequestration. Following this proposed, utilization of birch forests, our proposed approach for estimating stored carbon stocks can be used as an additional method for controlling the effectiveness of forest management measures.

**Author Contributions:** Conceptualization, N.F. and V.M.; methodology, N.F., I.B., A.K., V.M. and R.U.; software, N.F., I.B. and P.S.; validation, N.F., P.S., L.B., O.M. and S.Z.; formal analysis, I.B., I.T., R.G., O.M. and R.U.; investigation, N.F., P.S., I.B., I.T., R.G., A.K. and M.K.; data curation, N.F. and P.S.; writing—original draft preparation, N.F., I.B., S.Z. and L.B.; writing—review and editing, N.F., I.B., S.Z., V.M. and M.K.; visualization, I.T. and S.Z.; project administration, N.F. and P.S. All authors have read and agreed to the published version of the manuscript.

**Funding:** This research was performed within the state assignment framework of the Ministry of Science and Higher Education of the Russian Federation "Program for the creation and functioning of a carbon polygons on the territory Bashkortostan Republic "Eurasian carbon polygon" for 2022–2023 (Publication number: FEUR-2022-0001).

**Data Availability Statement:** The data presented in this study are available upon request from the corresponding author.

**Conflicts of Interest:** The authors declare no conflict of interest.

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
