# Peer review of "Estimation of Carbon Stocks of Birch Forests on Abandoned Arable Lands in the Cis-Ural Using Unmanned Aerial Vehicle-Mounted LiDAR Camera"

_forests, doi:10.3390/f14122392_

Round 1

Reviewer 1 Report

Comments and Suggestions for Authors

The author utilized LiDAR technology to examine the biomass information of abandoned birch forests and subsequently established the relationship between birch biomass information and carbon storage. This study is not only intriguing but also highly relevant to current mainstream research, making it a significant contribution to the field. The article boasts a rigorous conceptual framework and thorough theoretical analysis. However, several crucial aspects within the article remain unexplained. It is suggested that the author provide further elaboration on these points to enhance the readers' comprehension of the study.

1The article did not specify the laser model or operational process; please include this information.

2The process of point cloud processing and obtaining biomass information is not mentioned in the article; please provide further details.

3What is the rationale behind using the average chest diameter at a height of 1.3m above the ground in the study?

4The article refers to the use of the VLM series (Vilitek LLC, Moscow, Russia) cutting machine for tree crushing and carbon content determination. What is the sample size of the trees? Is it necessary to crush all trees?

5In the conclusion section, various parameters such as the sum of crown diameters multiplied by the average height are chosen as independent variables. Why were these specific variables selected as independent variables?

6The conclusion section mentions parameters such as canopy area. How were these values obtained?

7How was the non-linear function relationship in Table 1 determined? Why were other forms not considered? What was the original intention behind selecting this form?

8The correlation between the independent variables mentioned in Table 1 and biomass is high, with insignificant differences. What could be the reason for this result? Are all independent variables directly related to carbon content? Please provide an explanation from the author.

9How was the conversion coefficient mentioned in line 208 obtained? This parameter is crucial, yet it has not been thoroughly explained in the article, and the related references have not been found.

Author Response

The author utilized LiDAR technology to examine the biomass information of abandoned birch forests and subsequently established the relationship between birch biomass information and carbon storage. This study is not only intriguing but also highly relevant to current mainstream research, making it a significant contribution to the field. The article boasts a rigorous conceptual framework and thorough theoretical analysis. However, several crucial aspects within the article remain unexplained. It is suggested that the author provide further elaboration on these points to enhance the readers' comprehension of the study.

Answer: Dear Reviewer. We are thank you for careful review of our article. Your recommendations and comments are very useful and improved the quality of manuscript. Below are the answers to your comments (in the text of article corrections are marked in green color).

  1. The article did not specify the laser model or operational process; please include this information.

Answer: We agree with your opinion. Laser scanning of tree stands was conducted using a DJI Matrice 300 RTK UAV, equipped with a Zenmuse L1 LiDAR camera (SZ DJI Technology Co., Shenzhen, China). The Zenmuse Lidar L1 combines a Livox Lidar module, a high-precision IMU and a 1-inch CMOS camera on a 3-axis stabilised system (covering up to 2 km2 in a single span, vertical accuracy – 5 cm, horizontal accuracy – 10 cm. Point rate: 240.000 pts/s, detection range – 450 m). In the preparatory phase, a flight mission is planned – the survey area is set in KML format, which is loaded into the DJI Pilot 2 UAV's control panel software. After the flight plan is uploaded, the drone performs shooting in automatic mode. The images are saved in JPG format, and in parallel the georeferenced coordinates of the images are recorded using the D-RTK2 mobile station. The resulting images were imported into the DJI Terra program and stitched together automatically. 3D point cloud from LAS format with density from 300 to 950 pcs/m2 is formed. Post-processing of the stitched images was performed in LiDAR 360 Version 4.5 (Green Valley International, Berkeley, CA, USA). Necessary explanations are included in the text of manuscript (L. 139-148).

  1. The process of point cloud processing and obtaining biomass information is not mentioned in the article; please provide further details.

Answer: In the 1st stage, each UAV flyover section is cleared of points located outside the main scanning area ("outliers"). Next, the set of points is classified and divided into two types: points of the Earth's surface and other points above them. A digital elevation model (DEM) is built using the points of the first type. The remaining points are used to create a digital terrain model (DTM), which contains spatial information about the surface position of all objects of the areas above the Earth's surface. By excluding DEM data from the DEM, a digital forest canopy model (DFM) is created, which is an image of tree crowns. The pixel size of the resulting images is 4.0 cm. Necessary explanations are included in the text of manuscript (L. 150-158).

  1. What is the rationale behind using the average chest diameter at a height of 1.3m above the ground in the study?

Answer: The diameter at breast height (DBH) – 1.3 m height above the ground. In order to obtain the results of tree stand phytomass reserves, we were guided by the generally accepted methods of sampling plots and tree stand inventory (Hyyppa et al., 2020 a, b). Necessary explanations and references have been added to the manuscript (L. 127-132).

  1. The article refers to the use of the VLM series (Vilitek LLC, Moscow, Russia) cutting machine for tree crushing and carbon content determination. What is the sample size of the trees? Is it necessary to crush all trees?

Answer: On the sample plots, a complete enumeration of living trees and deadwood by thickness steps (2 cm) was carried out, followed by selection of an average model tree, from which age, air-dry mass of the trunk part, branches and leaves in the lower, middle and upper parts of the crown were determined, as well as sampling for analysis of their carbon content. That is, one model tree was selected for analysis within each sample plot. A total of 28 trees were sampled.

  1. In the conclusion section, various parameters such as the sum of crown diameters multiplied by the average height are chosen as independent variables. Why were these specific variables selected as independent variables?

Answer: The coefficients of the correlation (R) and determination (R2), and the estimation standard error (ESE) were used as regression models quality criteria. Further, the difference between the values of carbon stocks in the above-ground part of the stand calculated from LiDAR sur-vey data and the values obtained by the traditional method using a model trees was assessed. The best model for calculating the aboveground biomass of birch forests with the smallest difference between the results of traditional and LiDAR measurements was the model using multiplication of the trees amount by their average height. Therefore, it is this indicator that was chosen in this particular case.

  1. 6. The conclusion section mentions parameters such as canopy area. How were these values obtained?

Answer: The use of LiDAR data allows determining the crown areas of individual trees and the sum of tree crown areas per unit area. They are effective in estimating stand stocks in native forests.

  1. How was the non-linear function relationship in Table 1 determined? Why were other forms not considered? What was the original intention behind selecting this form?

Answer: Regression analysis was carried out in the program Statgraphics centurion XV. The «Comparison of Alternative Models» algorithm was used to select the optimal regression models. Linear and non-linear graphs were plotted for each dependence and the graph with the highest values of correlation coefficients (R) and determination coefficients (R2) was selected.

  1. The correlation between the independent variables mentioned in Table 1 and biomass is high, with insignificant differences. What could be the reason for this result? Are all independent variables directly related to carbon content? Please provide an explanation from the author.

Answer: In closed tree stands with high total crown cover (greater than 0.90), the mean relative deviations of UAV estimates of aboveground biomass from field measurements were close to 0 (Holiaka et al., 2021). Carbon content is independent of the metrics considered. Phytomass is dependent. The carbon content is a species-specific indicator – used to calculate carbon stocks per unit area.

  1. How was the conversion coefficient mentioned in line 208 obtained? This parameter is crucial, yet it has not been thoroughly explained in the article, and the related references have not been found.1. It is recommended to supplement the data of woody and herbaceous plants biomass.

Answer: Thanks for comment. The carbon content in the samples was determined using a CHNS EA-3100 elemental analyzer (Eurovector, Pavia, Italy). Conversion coefficients were calculated from the carbon content of the samples (fractional content) to calculate carbon stocks in stem wood and leaves. All results on carbon content of different biomass fractions are given in more detail in previous articles (Fedorov et al. 2023).

Fedorov, N.; Shirokikh, P.; Zhigunova, S.; Baisheva, E.; Tuktamyshev, I.; Bikbaev, I.; Komissarov, M.; Zaitsev, G.; Giniyatullin, R.; Gabbasova, I.; Urazgildin, R.; Kulagin, A.; Suleymanov, R.; Gabbasova, D.; Muldashev, A.; Maksyutov, Sh. Dynamics of biomass and carbon stocks during reforestation on abandoned agricultural lands in Southern Ural region. Agriculture. 2023, 13, 1427. https://doi.org/10.3390/agriculture13071427

Nikolai Fedorov

Ufa Institute of biology - Subdivision of the Ufa Federal Research Centre of the Russian Academy of Sciences

Prospect Octyabrya, 69, Ufa 450054, Russia

fedorov@anrb.ru

November 15, 2023

Reviewer 2 Report

Comments and Suggestions for Authors

The introduction should be slightly more extended to include information referring to the first part of the title. There are no numbers relating to carbon stocks of birch forests here.

Keywords should be analyzed again. Words that appear in the title should be avoided.

I have no objections to the methodology, results and discussion.

Conclusions should be improved. This part of the article cannot contain citations. Conclusions must relate directly to the results achieved.

The English language seems to need improvement. I don't know why, but the Authors use the word „message” instead of „article”.

Line 17. Instead of CO2 there should be CO2.

Lines 163 and 164. Instead of R2 it should be R2.

Figure 3b. On the vertical axis, please correct the unit.

Author Response

Dear Reviewer. We are thank you for careful review of our article. Your recommendations and comments are very useful and improved the quality of manuscript. Below are the answers to your comments (in the text of article corrections are marked in blue color).

The introduction should be slightly more extended to include information referring to the first part of the title. There are no numbers relating to carbon stocks of birch forests here.

Answer: We agree with your opinion. In the region, the average carbon stocks in 25–30-year old birch tree stand on abandoned agricultural land is 74.9 t/ha, which is close to that in Scandinavia for a 32-year old birch tree stand (87.7 t/ha). The comparison of our data with birch stands in other regions was discussed in more detail earlier in another article (Fedorov et al., 2023). Necessary explanations are included in the text of manuscript (L. 69-71).

Fedorov, N.; Shirokikh, P.; Zhigunova, S.; Baisheva, E.; Tuktamyshev, I.; Bikbaev, I.; Komissarov, M.; Zaitsev, G.; Giniyatullin, R.; Gabbasova, I.; Urazgildin, R.; Kulagin, A.; Suleymanov, R.; Gabbasova, D.; Muldashev, A.; Maksyutov, Sh. Dynamics of biomass and carbon stocks during reforestation on abandoned agricultural lands in Southern Ural region. Agriculture. 2023, 13, 1427. https://doi.org/10.3390/agriculture13071427

Keywords should be analyzed again. Words that appear in the title should be avoided.

Answer: Acknowledged. We changed the keywords. (L. 36, 37)

I have no objections to the methodology, results and discussion.

Conclusions should be improved. This part of the article cannot contain citations. Conclusions must relate directly to the results achieved.

Answer: We agree with your opinion. We moved a few sentences to the Discussion section and rewrote the Conclusion (L. 331-353)

The English language seems to need improvement. I don't know why, but the Authors use the word „message” instead of „article”.

Answer: Thanks for comments. Done (L. 21, 99).

Line 17. Instead of CO2 there should be CO2.

Answer: Done (L. 17).

Lines 163 and 164. Instead of R2 it should be R2.

Answer: Done (L. 170, 171).

Figure 3b. On the vertical axis, please correct the unit.

Answer: Thanks. We corrected and replaced Figure 3.

Nikolai Fedorov

Ufa Institute of biology - Subdivision of the Ufa Federal Research Centre of the Russian Academy of Sciences

Prospect Octyabrya, 69, Ufa 450054, Russia

fedorov@anrb.ru

November 15, 2023

Round 2

Reviewer 1 Report

Comments and Suggestions for Authors

The author provided detailed answers to the questions raised in the preliminary review, and some aspects have been improved. However, there are still some questions that the author did not directly address. We hope the author will carefully consider and revise the article accordingly.

In Table 2 of the article, the function models fitted show strong correlations, indicating that the parameters involved in the article will directly affect carbon content. Although the product of the sum of tree crown diameters and their average height has the highest correlation, the correlations of other parameters are also very high, and the difference is not significant. It is important to consider whether reconstruction errors might have an impact on the final parameter selection. Additionally, if multiple parameters are fitted together, it would be interesting to see if the correlation might be even higher.

We suggest that the author explore these questions further, as addressing them could potentially strengthen the article and provide more robust conclusions.

Author Response

The author provided detailed answers to the questions raised in the preliminary review, and some aspects have been improved. However, there are still some questions that the author did not directly address. We hope the author will carefully consider and revise the article accordingly.

Answer: Dear Reviewer. We are thank you for careful review of our article. Your recommendations and comments are very useful and improved the quality of manuscript. Below are the answers to your comments (in the text of article corrections are marked in green color).

In Table 2 of the article, the function models fitted show strong correlations, indicating that the parameters involved in the article will directly affect carbon content. Although the product of the sum of tree crown diameters and their average height has the highest correlation, the correlations of other parameters are also very high, and the difference is not significant. It is important to consider whether reconstruction errors might have an impact on the final parameter selection. Additionally, if multiple parameters are fitted together, it would be interesting to see if the correlation might be even higher. We suggest that the author explore these questions further, as addressing them could potentially strengthen the article and provide more robust conclusions.

Answer: Among of the five main forest stand parameters (trees amount, trees average height, sum of trees diameters, areas and volumes of tree crown), only the tree amount and its average height are independent. The sum of crown diameters and the sum of crown areas depend on the trees average height, which increases with age, and on trees amount. The sum of trees crown volumes also depends on the trees average height of trees as it is related to the age of the stand. Therefore, the three parameters (multiplying the sum of trees crown diameters by their average height, multiplying the sum of tree crown areas by their average height, multiplying the sum of tree crown volumes by their average height) may give an inaccurate estimate because the formula includes dependent variables. Thus, the multiplying the tree amount by their average height is the most appropriate. Calculation of multiple regression (by 3 or more parameters) with the introduction of additional dependent parameters into the formula will be incorrect. Necessary explanations are included in the text of manuscript (L. 288-298).

Nikolai Fedorov

Ufa Institute of biology - Subdivision of the Ufa Federal Research Centre of the Russian Academy of Sciences

Prospect Octyabrya, 69, Ufa 450054, Russia

fedorov@anrb.ru

November 15, 2023
